# Optical Properties and Microstructure of $TiN_xO_y$ and TiN Thin Films before and after Annealing at Different Conditions

**Hanan A. Abd El-Fattah** [1,*], **Iman S. El-Mahallawi** [1,2], **Mostafa H. Shazly** [3]
**and Waleed A. Khalifa** [1]

1   Metallurgical Engineering Department, Faculty of Engineering, Cairo University, Giza 12613, Egypt;
    ielmahallawi@Bue.edu.eg (I.S.E.-M.); waleed_khalifa@yahoo.com (W.A.K.)
2   Centre for Renewable Energy, The British University in Egypt BUE, Cairo 11873, Egypt
3   Mechanical Engineering Department, Faculty of Engineering, The British University in Egypt,
    Al-Shorouk City, Cairo 11873, Egypt; Mostafa.Shazly@Bue.edu.eg
*   Correspondence: enghanann@gmail.com; Tel.: +20-10-0673-1079

**Abstract:** TiN and $TiN_xO_y$ thin films share many properties such as electrical and optical properties. In this work, a comparison is conducted between TiN (with and without annealing at 400 °C in air and vacuum) and $TiN_xO_y$ thin films deposited by using RF magnetron sputtering with the same pure titanium target, Argon (Ar) flow rate, nitrogen flow rates, and deposition time on stainless steel substrates. In the case of $TiN_xO_y$ thin film, oxygen was pumped in addition. The optical properties of the thin films were characterized by spectrophotometer, and Fourier transform infrared spectroscopy (FTIR). The morphology, topography, and structure were studied by scanning electron microscope (SEM), atomic force microscope (AFM), and X-ray diffraction (XRD). The results show that both thin films have metal-like behavior with some similarities in phases, structure, and microstructure and differences in optical absorbance. It is shown that the absorbance of TiN (after vacuum-annealing) and $TiN_xO_y$ have close absorbance percentages at the visible range of light with an unstable profile, while after air-annealing the optical absorbance of TiN exceeds that of $TiN_xO_y$. This work introduces annealed TiN thin films as a candidate solar selective absorber at high-temperature applications alternatively to $TiN_xO_y$.

**Keywords:** selective absorber; PVD sputtering; optical properties; microstructure; TiN; $TiN_xO_y$

## 1. Introduction

Direct conversion of solar radiation into thermal energy, such as in the cases of solar water heater and concentrated solar power (CSP) plant, has high potential in hot regions. CSP has some unique advantages such as higher thermal energy storage capability and higher energy-conversion efficiency. In a CSP plant, the solar selective absorber (SSA) material is the key element, which must have high efficiency to absorb solar energy with minimum losses due to black body emission [1]. An SSA consists of several layers to achieve selectivity, which are an infrared (IR) reflective layer with high reflectance in IR regions (usually a metallic layer), solar absorbing material in visible and near-IR region (usually cermets or $TiN_xO_y$), and an antireflection (AR) layer on the top to restrict the solar radiation (usually nitrides or oxides). Over the recent decades, the development of selective solar absorbers with absorbance higher than 95% and emittance lower than 1% represented the main goal for many researchers, e.g., [2].

Different types of SSA, such as metal–dielectric composite, multilayer absorbers and semiconductor–metal tandems have been developed due to their higher thermal absorbance, higher

thermal stability and low IR emittance [3–6]. In recent years, many types of selective solar absorber coatings for mid-to-high-temperature applications were studied, but only a few of them have been successfully commercialized. Among these films, $TiN_xO_y$ have captured the interest as an absorber layer in SSA applications for high temperatures [7]. $TiN_xO_y$ is classified as a cermet coating deposited on different substrates followed by an AR layer made of fused quartz ($SiO_2$) [7].

Many researchers deposited $TiN_xO_y$ using reactive magnetron sputtering and by changing the (N/O) ratio, with different types of AR layers such as $TiO_2/Si_3 N_4/SiO_2$ and $TiO_2/SiO_2$ to obtain efficient SSA [3,4,8]. Chen et al. [4] found that the behavior of $TiN_xO_y$ films was changed from metallic to dielectric behavior by increasing the oxygen content. In the case of low N/O ratio, the formed $TiN_xO_y$ was transparent in the solar spectrum region, while high N/O ratio resulted in metallic behavior and high-reflection $TiN_xO_y$ thin films [4]. Zhang et al. [3] determined the optimum N/O flow rate, which resulted in a solar selective absorber with a high absorbance of 97.55%. Kazemeini et al. [8] used DC sputtering reactive process to produce $TiN_xO_y$ film by using Ti target and $O_2/N_2$ ratio of 1/4. X-ray diffraction studies showed that the produced film forms a crystalline structure and has the main peaks of (111), (200), and (220). These films appear in a golden color similar to TiN. F. Vaz et al. [9] reported the preparation of $TiN_xO_y$magnetron sputtered thin films, where the color of deposited $TiN_xO_y$films varied from the glossy golden one for low oxygen content (characteristic of TiN films) to dark blue color for higher oxygen contents.

In spite of the advancements in studying and characterizing $TiN_xO_y$ as a selective absorber, it is not suited for high temperature applications above 400 °C, while TiN is reported to be stable till 500 °C [10]. In addition, the fabrication process is relatively complicated and expensive. Alternatively, Titanium Nitride (TiN) films, which share $TiN_xO_y$ many properties such as the dielectric properties can be used a candidate material for SSA applications. TiN, the most popular transition metal nitride, has a golden color and is characterized as an extremely hard material, with a high melting point (2950 °C), high thermal and chemical stability at elevated temperatures, and ease to manufacture as compared to $TiN_xO_y$ [11,12]. Several studies investigated the structure evolution and electrical properties of TiN thin films produced by different deposition parameters [13–20]. Several researchers have suggested [10,21–24] that TiN may be used as an absorber layer for high-temperature applications; however, limited work appears in literature on the optical properties of TiN as a selective absorber [21–24]. $TiN_xO_y$ had captured attention in the last decades, although TiN has almost the same properties, as will be proven in this work in addition to other advantages regarding its stability at higher temperatures and ease of preparation.

In this work, $TiN_xO_y$ and TiN thin films were deposited by reactive magnetron sputtering using the same process conditions, and the TiN thin films were annealed at 400 °C (in air and in vacuum). The aim of this work is to compare and correlate the optical properties and microstructure of as-deposited and annealed thin films. The comparison is based on deposited thin films only without any AR layers to determine the merit of the thin film alone. The comparison made in this work between TiN and $TiN_xO_y$ aims at providing further evidence for applicability of TiN instead of $TiN_xO_y$ as a selective absorber. $TiN_xO_y$ and TiN thin films are deposited on Stainless Steel (304L) substrates (SS) with the same titanium target, the same Ar gas flow rate, and the same working pressure. Only $N_2$ gas was used for depositing the TiN thin film, whereas $N_2$ and $O_2$ were used for depositing the $TiN_xO_y$ thin film. SS substrate is used because it is considered as the best candidate material for CSP tube fabrication for high-temperature applications. The $N_2/O_2$ flow rate used in this work is chosen based on reviewing previous works [3,4] and initial deposition trials. Post-deposition vacuum annealing is adopted in this work as an option for residual stress relief and energy reduction of the sputtered thin films. The structure of the deposited thin films is studied by scanning electron microscopy (SEM), atomic force microscopy (AFM), and X-ray diffraction (XRD). The optical properties are studied using spectrophotometer and Fourier Transform Infrared (FTIR) spectroscopy.

## 2. Experimental

### 2.1. Thin Film Fabrication

$TiN_xO_y$ and TiN thin films were deposited on stainless steel (304L) substrates by Protec Nano-Flex 400 RF Magnetron (Bedizzola, Italy) sputtering tool at a frequency of 13.6 MHz and maximum output power of 2.5 kW. Pure titanium target (99.999%) with size ($30 \times 10$ cm$^2$) was used for the deposition of all thin films in reactive medium using Ar, $O_2$, and $N_2$ gases. The SS substrates had dimensions $4 \times 4$ cm$^2$. Five substrates (including one for measuring the thickness of the deposited layer) were introduced into the chamber for each run. All substrates were cleaned in acetone and isopropanol, followed by nitrogen drying. The cleaned and dried substrates were fixed in the deposition chamber with bias voltage equal to 150 V and 3 rpm rotation velocity. During the deposition, sputtering power was maintained at 1.15 kW, and the chamber was pumped down to $10^{-4}$ Pa (the base pressure). The chamber has a circular geometry with diameter 50 cm. The chamber temperature reached 160 °C during the sputtering process, and the distance between the target and the substrate is equal to 10 cm. The chamber temperature reached 160 °C during the sputtering process as recorded on the digital screen.

A thin layer of metallic Ti was deposited initially in all runs at different conditions, using only Ar with 200 sccm flow rate for 10 min. This step assisted in realizing vacuum conditions during the TiN deposition, as well as the elimination of oxygen traces inside the chamber by reaction with Ti [25]. This was followed by the deposition of another layer of TiN or $TiN_xO_y$ thin film at a constant deposition pressure of 1 Pa and the same Ar flow rate (200 sccm). The $TiN_xO_y$ thin film was deposited by using $N_2$, $O_2$ gases with flow rates 70, 20 sccm (corresponding to a ratio of 3.5/1), respectively, for 10 min. The TiN thin film was deposited by using only $N_2$ gas with flow rate 70 sccm for 10 min (corresponding to $N_2$% age of 26%). It should be noted here that $O_2$ and $N_2$ flow rates used in the present work are different from previously reported work due to different chamber sizes used; however, the ratio $O_2/N_2$ is selected based on the results obtained in previous work [3,4,8].

### 2.2. Annealing of TiN Thin Films

The TiN thin film deposited in this work was subjected to further annealing in ambient air and in vacuum in an OTF-1200X-II-UL (MTI Corporation, Richmond, CA, USA) furnace. This furnace is a split three-zone tube furnace, which can achieve faster heating up to 1200 °C. TiN thin films were annealed at 400 °C for 2 h in air and in vacuum.

### 2.3. Thin Film Characterization

The absorbance and emittance of the thin films were determined by measuring the reflectance at room temperature. The reflectance was measured by using spectrophotometry and FTIR in the visible and IR ranges. Shimadzu UV-3600 (Kyoto, Japan) spectrophotometry was used in this work for the measurements taken in wavelength range between 200 and 2500 nm. FTIR spectrometer JASCO FT/IR-4100 (Easton, MD, USA) was used for long wavelength (2.5–25 μm) measurements.

According to Kirchoff's law, the absorbance $\alpha(\lambda)$ is equal to the thermal emittance $\varepsilon(\lambda)$ and for opaque materials both can be expressed to the total hemispherical reflectance $R(\lambda)$ [26] by the following equation:

$$\alpha(\lambda) = \varepsilon(\lambda) = 1 - R(\lambda) \tag{1}$$

The thermal emittance $\varepsilon(T)$ at given temperature $T$ can be calculated by Equation (2) [27]:

$$\varepsilon(T) = \frac{\int_{\lambda_{min}}^{\lambda_{max}} [1 - R(\lambda, T)] B(\lambda, T) \mathrm{d}\lambda}{\sigma T^4} \tag{2}$$

where $\sigma$ is the Stefan–Boltzmann constant ($5.6696 \times 10^{-8}$ W m$^{-2}$ K$^{-4}$) and $B(\lambda,T)$ is the spectral irradiance of a blackbody at the wave length $\lambda$ and temperature $T$ which is given by Planck's law in Equation (3):

$$(\lambda, T) \;=\; \frac{c_1}{\lambda^5 [e^{(c_2/\lambda T)} - 1]} \tag{3}$$

where $c_1 = 3.7405 \times 10^8$ W $\mu m^4$ $m^{-2}$ and $c_2 = 1.43879 \times 10^4$ $\mu m$ K are Planck's first and second radiation constants.

Scanning Electron Microscopy (SEM) model Quanta 250 FEG (Field Emission Gun, Thermo Fisher Scientific, Hillsboro, OR, USA) attached with EDX Unit (Energy Dispersive X-ray Analyses), with accelerating voltage 30 kV, and high magnification range was used to investigate the thin films microstructure.

A Keysight 5600LS (Santa Rosa, CA, USA) Atomic Force Microscope (AFM) was used to measure the surface topography of the deposited TiN thin films before and after annealing. The roughness and thickness of the deposited thin films were measured using KLA Tencor P-17 stylus profiler (Milpitas, California, USA) with standard surface measurement tool and a step height repeatability of 4 Å, one-sigma or better on samples up to 1 $\mu m$ tall.

Structural characterization of the thin film was carried out using X-ray diffraction (XRD), manufactured by Panalytical B.V. Co., Almelo, The Netherlands. The X-ray source was a Cu target operating at 40 kV and 30 mA with a continuous scanning type. The geometry configuration at 2θ of the incident angle was used to determine the crystallographic phases and the structure of the thin films.

## 3. Results and Discussion

### 3.1. SEM Study and Surface Topography

Despite the significance of the surface morphology and the microstructure features to the physical properties, most of the published work on optical and electrical properties of TiN thin films lacks microstructure studies. Figure 1 shows the SEM micrographs of the as-deposited TiN, $TiN_xO_y$ and annealed TiN at 400 °C in air and in vacuum thin films. This figure shows that the as-deposited microstructure exhibits well-defined large areas conforming to the original substrate grains with clear facets separated with large surface cavities along substrate original boundaries. Figure 1a shows that the average island size of the TiN thin film is approximately 8 $\mu m$, while Figure 1c reveals that the island size of $TiN_xO_y$ thin film is larger and almost equals to 12 $\mu m$. The large size of the islands and their contrast with the original substrate grain size (10 $\mu m$) suggests epitaxial growth relationship [28]. The TiN thin film has large valleys and the valleys between the islands are larger than that of $TiN_xO_y$ thin film.

It can also be revealed from Figure 1c,d that the island size of TiN after annealing in air and in vacuum at 400 °C is approximately 9 and 8 $\mu m$, respectively. There is no difference in crystalline structure after annealing in both cases. However, slight growth occurred in the case of annealing in air, probably due to the fact that, after annealing, the size of valleys were decreased.

Figure 1 also presents the AFM images of the TiN, $TiN_xO_y$, annealed TiN in vacuum and in air thin films. The thicknesses of the two thin films were measured and found to be 94 and 103 nm for TiN and $TiN_xO_y$, respectively. According to stylus measurements, the surface roughness of the substrate was 150 nm, the surface roughness of TiN thin film is 110 nm, whereas the $TiN_xO_y$ surface roughness is 120 nm. After annealing, the surface roughness of the annealed TiN in air and vacuum changed to 90 nm, and 100 nm, respectively, the reduction in surface roughness after deposition is explained by filling the peaks and valleys of the substrate. The thin film roughness measured by AFM was found to be 20.6, 2.15, 2.44, and 1.82 nm; for the $TiN_xO_y$, TiN as-deposited, after air annealing and vacuum annealing, respectively, suggesting an effect induced by different microstructures. As shown in Figure 1, the surface topography profile is different for both TiN and $TiN_xO_y$ thin films. The structure of $TiN_xO_y$ complies with previous reports showing that $TiN_xO_y$ thin film coatings with lower and medium percentage of oxygen illustrated crystalline structure and bright yellow pink color, unlike films with high oxygen content, which were reported to be amorphous, and having dark blue color [9].

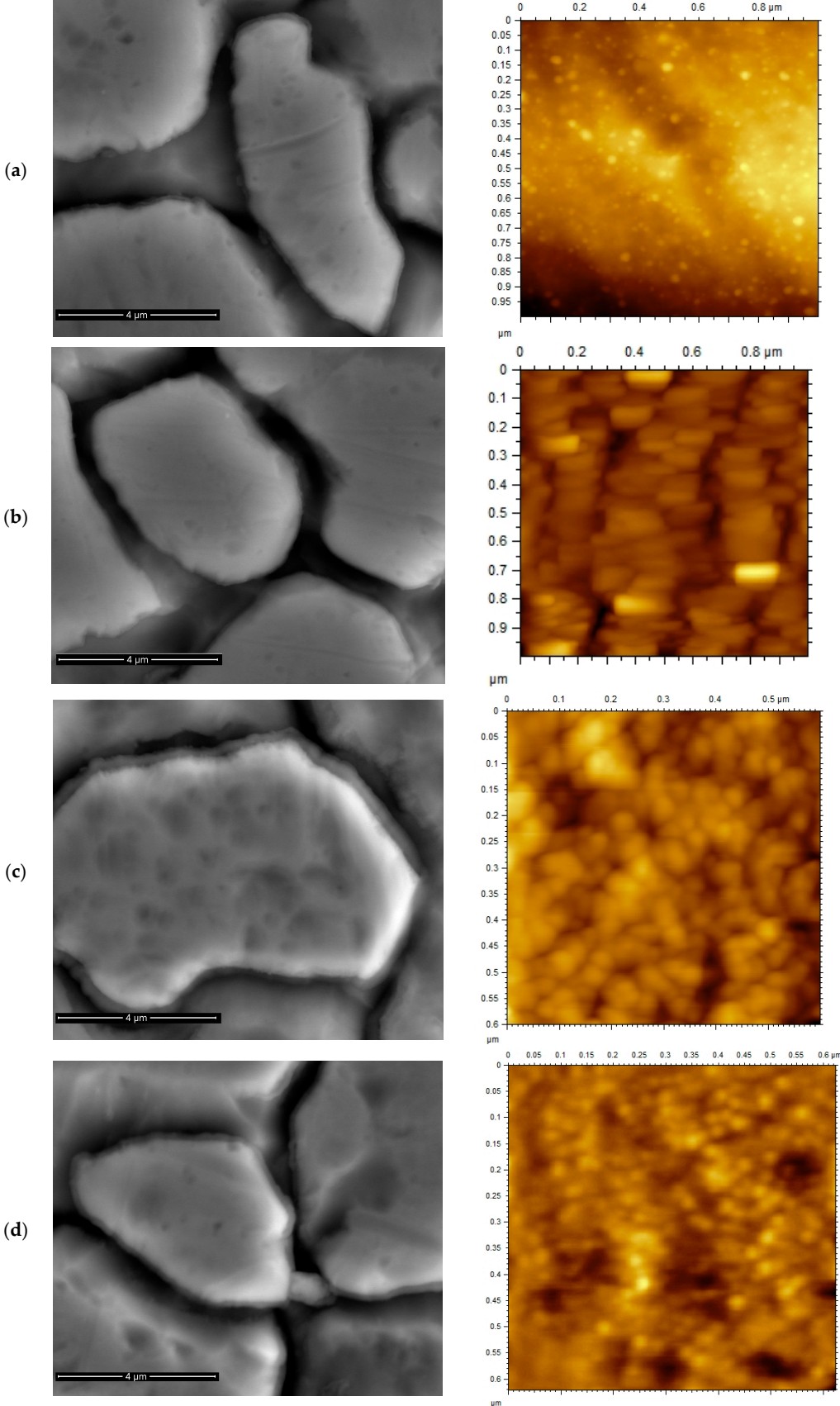

**Figure 1.** SEM and AFM images of (**a**) TiN thin film, (**b**) TiN$_x$O$_y$ thin film, (**c**) TiN after annealing at 400 °C in air, and (**d**) TiN after annealing at 400 °C in vacuum.

The morphology of the deposited nanoparticles is also revealed from Figure 1, showing the AFM images. Nanometric-sized particles of TiN are clearly identified in Figure 1a, and globular well-defined equiaxed nanoparticles are shown in Figure 1c,d. Annealing in vacuum caused the formation of equiaxed nanoparticles, whereas annealing in air (Figure 1c) has coarsened the nanoparticles and made them more homogenous and equiaxed. The presence of dense $TiN_xO_y$ layers of columnar nanoparticles is observed in Figure 1b. Similar microstructures were reported by few works [29].

The nanoparticles size was obtained from AFM images shown in Figure 1. The nanoparticles size measurements from AFM images revealed that the nanoparticles size of TiN crystallites was 25 nm and the nanoparticles size of the $TiN_xO_y$ columnar nanoparticles was 150 nm width × 20 nm thickness. The size of the TiN equiaxed nanoparticles annealed at 400 °C in air remained 25 nm and dropped slightly to 18 nm for the annealed at 400 °C in vacuum condition. The change in the morphology and dimensions of the nanoparticles during annealing has probably caused the increase in the absorbance of the annealed TiN in air and in vacuum, as will be shown later.

Figure 2 shows the EDX analysis of TiN, $TiN_xO_y$, annealed TiN in air and in vacuum thin films. Figure 2a shows the presence of Ti and $N_2$ in the TiN thin film and the absence of $O_2$. The EDX analysis shows the trend for N/Ti in both TiN as deposited and as annealed in vacuum as shown in Table 1. It is shown from Table 1 that the deposited thin film consists of both stoichiometric and overstoichiometric TiN structures, where the overstoichiometric TiN is deficient in Ti. Figure 2b shows the presence of Ti, $N_2$, and $O_2$ in the $TiN_xO_y$ thin film. The high peaks of iron (Fe), Ni, and Cr are reflected from the substrate. Though EDX analysis is not capable of measuring the composition of the coating layer, the data are just used for trends, from which it is shown that the percentage of Ti in both TiN and $TiN_xO_y$ thin films is very close. The difference in chemical composition of the two thin films confirms that the $TiN_xO_y$ thin film is successfully deposited, and that the deposited layer is possibly stoichiometric $TiN_xO_y$. The $N_2$ percentage is reduced in the $TiN_xO_y$ and the air-annealed TiN thin films, combined with the appearance of an $O_2$ peak. It is worth observing the similarities in EDX analysis of the $TiN_xO_y$ and the air-annealed TiN, though their microstructures are different as shown in Figure 1b,c. Their optical performance will be shown later to be also different.

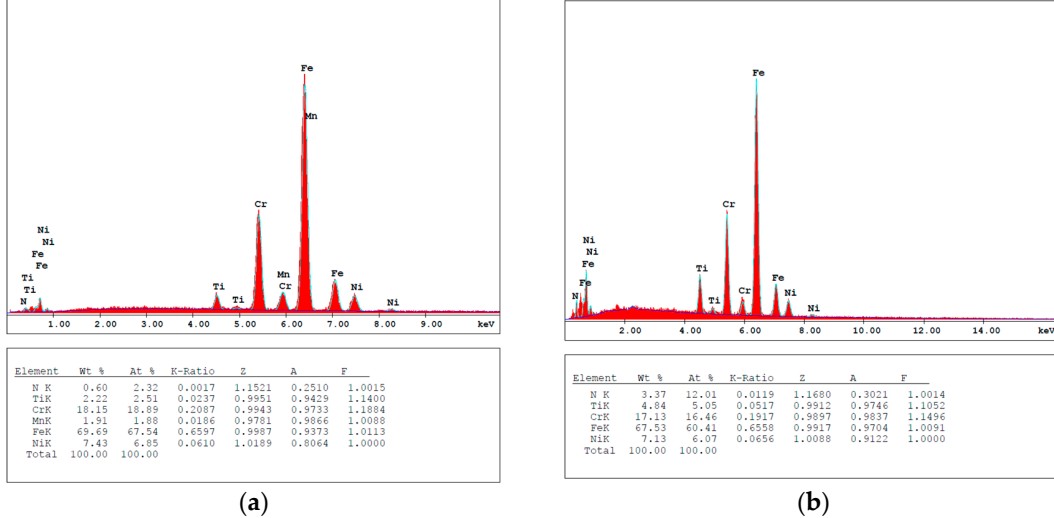

(**a**) (**b**)

**Figure 2.** *Cont.*

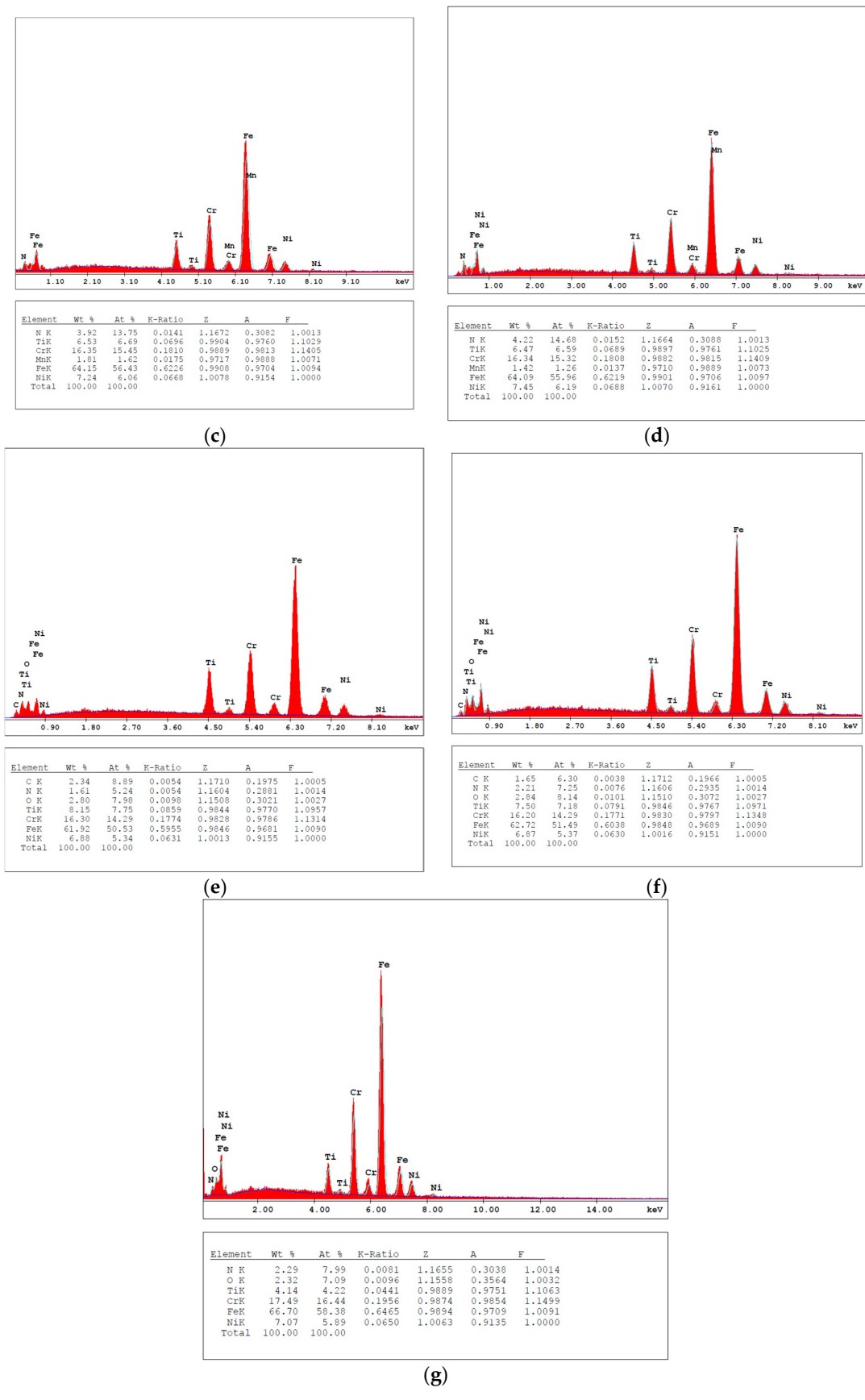

**Figure 2.** EDX of TiN (**a**,**b**), annealed TiN at 400 °C in vacuum (**c**,**d**), annealed TiN at 400 °C in air (**e**,**f**), and TiN$_x$O$_y$ (**g**).

**Table 1.** Composition of TiN, TiN$_x$O$_y$, and annealed TiN at 400 °C in air and in vacuum thin films [12,28].

| Condition | at % Ti | wt % Ti | at % N | wt % N | at % O | wt % O | N/Ti% | N at % |
|---|---|---|---|---|---|---|---|---|
| As deposited TiN | 2.51 | 2.22 | 2.32 | 0.6 | – | – | – | 2.32/4.83 = 48% |
| As deposited TiN | 5.05 | 4.84 | 12.01 | 3.37 | – | – | 3.37/4.84 = 1/1.43 | 12.01/17.06 = 70% |
| Annealed TiN in vacuum | 6.59 | 6.47 | 14.68 | 4.22 | – | – | 4.22/6.47 = 1/1.5 | 14.68/21.27 = 69% |
| Annealed TiN in vacuum | 6.69 | 6.53 | 13.75 | 3.92 | – | – | 3.92/6.53 = 1/1.66 | 13.75/20.44 = 67% |
| Average of TiN annealed in vacuum | 6.64 | 6.50 | 14.22 | 4.07 | – | – | – | – |
| Annealed TiN in air | 7.18 | 7.5 | 7.25 | 2.21 | 8.14 | 2.84 | – | – |
| Annealed TiN in air | 7.75 | 8.15 | 5.24 | 1.61 | 7.98 | 2.8 | – | – |
| Average of TiN annealed in air | 7.47 | 7.83 | 6.25 | 1.91 | 8.06 | 2.82 | – | – |
| As deposited TiN$_x$O$_y$ | 4.22 | 4.14 | 7.99 | 2.29 | 7.09 | 2.32 | – | – |

The TiN stoichiometry relationships are explained [12], according to the TiN phase diagram, that by increasing the nitrogen concentration, the titanium lattice expands in order to accommodate dissolved nitrogen atoms; the maximum concentration being 23 at % at 1050 °C, corresponding to a ratio N/Ti = 0.3. For nitrogen concentrations of 33 at %, and temperatures lower than 1050 °C, the Ti$_2$N phase with a body center tetragonal structure can be observed. For nitrogen concentrations around 37.5%, and temperatures lower than 800 °C, the existence of Ti$_2$N($\alpha$) or $\delta'$-Ti$_2$N phases is thermodynamically possible. The TiN phase, which has a face-centered cubic structure (FCC), also known as $\delta$-TiN, can be found with nonstoichiometric compositions, TiN$_x$, in the wide nitrogen compositional range of 40–55 at %. Based on XRD and XPS, it has been shown that the presence of oxygen impurity concentrations in TiN sputtered at low pressures or nitrogen flow rates results in inconsistency in stoichiometry before and after annealing [25] due to oxygen replacement. However, on the contrary, Yin et al. [22] showed that TiN produced at a higher nitrogen pressure could, also, absorb oxygen more easily into its bulk and was less oxidation-resistant during heat treatment [22]. However, based on XRD and XPS, it was shown [25] that the ratio of Ti concentration to that of N is 0.96–1.04 for all N$_2$ flow rates in the range (20% to 95%). Based on the N at % reported in Table 1, the formed TiN in this work is deficient in Ti as the N at % is higher than 50%. Overstoichiometric TiN was reported [25] for N/(N + Ar) ratios higher than 20% (in this work this ratio is 26%).

## 3.2. XRD Analysis

Figure 3 shows the XRD of both TiN, TiN$_x$O$_y$, annealed TiN in air and in vacuum thin films, from which it is shown that all conditions show signals at almost the same angles. The XRD patterns reveal that TiN and TiN$_x$O$_y$ thin films exhibit crystalline structures. The presence of three peaks are marked out on the XRD pattern for all thin films as shown in Figure 3. These peaks correspond to the (111), (200), and (220) planes, respectively, which were reported for TiN thin films [30] and TiN$_x$O$_y$ thin films [8], suggesting some overlapping of the signals with those obtained from the substrate (111). The SS substrate showed (111) peak at angle 2θ = 44.49. Different angles of TiN peaks are reported in literature [12,16,24,29] (as shown in Table 2), this is probably due to the different stichometry of TiN structure obtained from different manufacturing techniques of thin film. The lattice parameter of the as-deposited TiN with an austenitic microstructure was calculated by Bragg's Law for the (111), (200), and (220) peaks and was found to be 3.6 Å [29]. In the lattice structure, Ti sites are completely filled and vacancies exist only on the N sites. However, several research works [12,25,29] have illustrated the formation of nonstoichiometric (sub or over) TiN$_x$, which exhibits deficient Ti atoms and has superior properties. This suggests that the structure is TiN$_x$ [25]. It is also noted from Table 1 that an increase in the Ti at % occurs after annealing the TiN thin film. This suggests reducing defects as defects are associated with N sites in TiN.

Annealing was also shown as an additional postdeposition treatment, which can be used to optimize the properties and reduce the stresses and lattice defects induced during deposition. However, the literature contained contradictory reports about the effect of annealing on electrical and optical properties [16,24]. Popovic et al. [16] annealed deposited TiN thin films at 600 °C and 700 °C in nitrogen ambient and vacuum furnace, and it was found that the annealing process resulted in a decrease in the absorption coefficient and the optical properties of the TiN thin film, compared to the as-deposited state. Gao et al. [24] proposed heating the substrate for annealing to 300 °C before depositing the TiN thin film and it was found that after heating the substrate, the thin film exhibits an absorbance of about 0.92 and an emittance equal to 0.11.

The most predominant peak was (200) for all TiN samples and was (111) for the $TiN_xO_y$. The $N_2$ flow rate in this work is 26%; it was shown [25], based on XRD and XPS, that at various nitrogen pressures TiN phases are deposited, and as $N_2$ flow rates increase above 20%, the deposited TiN shows orientation and a predominant peak for (200) orientation in favor of the (111) orientation [25]. $TiN_xO_y$ thin films were reported to exhibit crystalline structures at moderate $O_2$ levels, whereas increased $O_2$ levels were shown to promote significant loss of crystallinity and even amorphous structures [9,16]. The produced $TiN_xO_y$ thin film in this work was deposited at $O_2$:$N_2$ ratios of 1:3.5 and exhibited a dark golden yellow color and a crystalline structure.

**Table 2.** XRD angles.

| Reference | Phase | Angle (2θ) |
|-----------|-------|------------|
| This work | (111), (200), (220) | 43.03, 51.6, 74.46 |
| [24] | (111), (200), (220) | 43, 51.5, 74.5 |
| [29] | (111), (200), (220) | 43, 51.5, 74.5 |
| [16] | (111), (200) | 36.2, 42.2 |
| [12] | (111), (200), (222) | 36.5, 43, 77 |

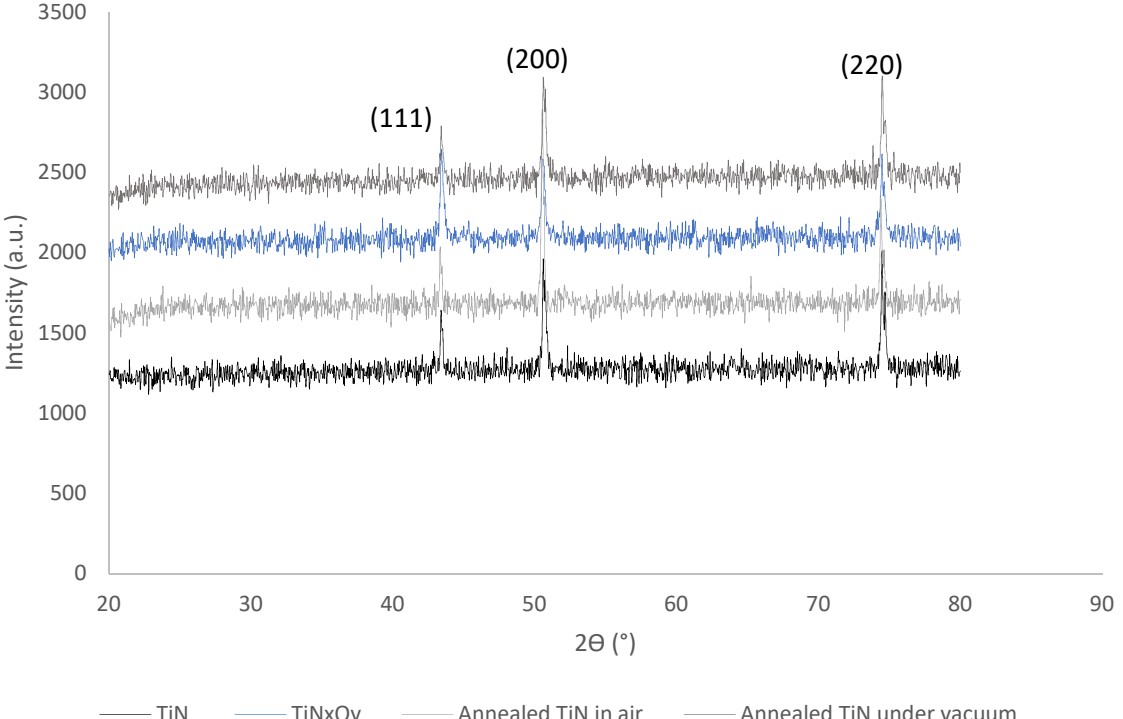

**Figure 3.** XRD of TiN, $TiN_xO_y$, annealed TiN in air, and annealed TiN in vacuum.



The values of the mean grain size (crystallite size) and microstrain obtained for the as-deposited and annealed thin films are summarized in Table 3. The broadening of the 111, 200, and 220 peaks was used to calculate the average crystallite size using Scherrer's equation. It was found that the crystallite size of TiN thin film was 4 nm and the crystallite size of $TiN_xO_y$ thin film was 3.5 nm. The crystallite size for the TiN film annealed at 400 °C in air was 6 nm and increased to 7 nm in the case of the film annealed at 400 °C in vacuum. Light scattering and trapping at the junctions of tiny microcrystalline grains can affect the absorbance. Hence, achieving a fine grain size is extremely important to achieve good absorbance. The broadening of XRD peaks in nanocrystals is common and can also occur due to strain in the crystal or overlapping with other peaks. Peak broadening is normally seen in crystallites smaller than 100 nm.

**Table 3.** Grain size and strain of as-deposited TiN, annealed TiN in air and in vacuum, and $TiN_xO_y$ thin films.

| Condition | $D$ (nm) | $\acute{\eta}$ ($\times 10^3$) |
|---|---|---|
| TiN | 4 | 6.4 |
| Annealed TiN in Air | 6 | 4.8 |
| Annealed TiN in Vacuum | 7 | 4.55 |
| $TiN_xO_y$ | 3.5 | 6.5 |

The XRD results obtained in this work show that though the postdeposition annealing of the films did not cause significant variation in stoichiometry, it affected the structural parameters such as microstrain and crystallite size in a manner showing the effect of annealing on reducing the lattice strain and the defects resulting from the deposition conditions. This is also supported by the slight increase in Ti at % after vacuum annealing (as shown in Table 1) and the decrease in microstrain after annealing as shown in Table 3.

### 3.3. Optical Properties

The absorbance of the deposited and annealed thin films is given in Figure 4. Figure 4 indicates that the TiN thin film has high IR reflectance and a relative steep edge in the visible region, which is a well-known, metal-like behavior. The relative steep edge refers to interband transitions involving the d-type free electrons, which means it contains conduction electrons resulting in metal-like electrical conductivity [31]. At higher incident energies (around wavelength 450 nm), interband transitions, which increase the number of nitrogen p-states and titanium d-states, take place resulting in the specific shape of the TiN absorbance pattern or reflectance pattern [31]. The saturation effect takes place around 850 nm with a plateau. As shown in Figure 4, the absorbance of the TiN layer increases from approximately 40% to 90% and then decreases again to reach approximately 30% due to interbands and transition behavior of TiN.

At higher incident energies (around wavelength 570 and 700 nm), interband transitions take place in case of the annealed TiN at 400 °C in air for 2 h, which increase the number of nitrogen p-states and titanium d-states, resulting in the specific shape of the absorbance pattern or reflectance pattern [31]. In the case of annealing in vacuum around 500 nm, interband transitions occurred. The annealed TiN sample in both cases has the same pattern of reflectance of the as-deposited TiN thin films, but the absorbance increases to more than 95% (air annealing) and 92% (vacuum annealing) and the interband transitions occur around higher wavelengths. The saturation effect takes place here at different wavelengths from the deposited TiN thin films with a plateau. As shown in Figure 4, the absorbance of the annealed TiN in air increases from approximately 65% to 95% and then decreases again to reach approximately 30% due to interbands and transition behavior. The absorbance of annealed TiN in vacuum increases from approximately 50% to 92% and then decreases again to reach approximately 30% due to interbands and transition behavior. The role of microstructure and morphology of the

nano-deposited grains has not been discussed before. This work shows that the change in the TiN grains morphology after annealing causes an effect on the optical properties. Though, this could be attributed to the effect of grain size and morphology on the surface roughness of the thin film.

Similarly, $TiN_xO_y$ film shows metallic-like behavior as the TiN films. A steep edge appears around 450 nm and saturation takes place around 850 nm with a peak absorbance value of approximately 94%. The absorbance of $TiN_xO_y$ is explained in terms of composite structures of dielectric materials dispersed with particle-formed conductor. In this case, TiN plays the major role as the absorbance center due to its ability to induce the interband transitions. With increasing the amount of TiN, the absorbance and optical properties increase; however, emittance also increases [32]. The change in optical properties of $TiN_xO_y$ films is intrinsically due to the change of electronic structure dominated by the N/O ratio. Intraband and interband electronic transitions occurring during the interaction of material with incident light determines the optical properties of a $TiN_xO_y$ film [33]. The optical properties can be designed if the density of these free electrons can be varied systematically via changes in the stoichiometry parameter [33]. With the incorporation of oxygen, the electronic structure can be systematically varied, resulting in the adjustable optical and electrical properties of $TiN_xO_y$ films. The optical properties of $TiN_xO_y$ films with high N/O ratio are dominated by the intraband transition from free electrons and behave like metallic films. With increasing oxygen content, the contribution of the interband transitions, due to bound electrons, became more pronounced, while the contribution of the intraband transitions is less important [34]. When $TiN_xO_y$ films change to be oxygen-rich, the optical properties are governed by the interband transitions, while the band gap will increase with the $O_2$ concentration [35], which are characteristic of semiconducting and insulating materials.

The $TiN_xO_y$ thin film deposited in this work behaves in optical properties as TiN thin film. Higher optical properties of $TiN_xO_y$ thin films are reported in literature [3], probably due to the AR layer deposited above the $TiN_xO_y$. As mentioned before, different types of AR layers such as $TiO_2/Si_3N_4/SiO_2$ and $TiO_2/SiO_2$ are deposited above $TiN_xO_y$ [3,4,8]. The problem with TiN is that, above 450 °C, it is oxidized and forms a thin 25 nm thickness sublayer of titanium oxide on top of its surface [22]. Oxidation is motivated by the high inherent energy inside the deposited layers, annealing acts to relieve the stresses and reduces the energy for oxidation, afterwards during service. The presence of an AR layer will protect the TiN from oxidation and improve its optical properties. Gao et al. [24] studied $Al_2O_3$ as AR layer for TiN selective absorber and got good absorbance with lower emittance. $Al_2O_3$ is a simply obtained AR and is easily deposited compared to the complicated AR layers used above $TiN_xO_y$ and it has the same function.

Figure 5 shows the emittance of TiN, $TiN_xO_y$, annealed TiN in air and annealed TiN in vacuum. The emittance of TiN and $TiN_xO_y$ thin films is almost the same. The values of emittance are compatible with the absorbance values. Emittance of annealed TiN in both cases is enhanced. The best emittance of all is shown to belong to the annealed TiN in vacuum, as it reached values below 1%.

TiN thin films had been successfully used in many industrial applications, ranging from thin films for electronic applications to thicker films for cutting tools. Yet, despite its high melting point, stability at elevated temperatures, and good optical absorbance (almost similar to $TiN_xO_y$ thin films), its function as a selective absorber is questionable. The main reason for this is its readiness to absorb oxygen easily during exposure to air at relatively high temperatures. However, research and industry are working to develop easy and cost-effective TiN thin films for solar selective coatings with a high absorbance and low emittance.

The conditions realizing high solar absorbance and low thermal emittance are such that the reflectance should be low in the solar radiation range and high in the infrared range. The transition point between those two regions is at the cross-point of the solar radiation spectrum and the blackbody radiation spectrum. In the mid-temperature SSA application (e.g., 400 °C), the blackbody radiation exceeds the solar radiation at the wavelength of 1.5 μm [10]. On the other hand, around 90% of solar energy is in the wavelength range of 0.3–1.5 μm [28]. Therefore, the transition point ($\lambda_c$) is usually set at the wavelength of 1.5 μm [3].

Previous work has shown that the properties of the thin films, such as preferred orientation of lattice plane, optical and electrical properties, etc. are highly dependent on the deposition parameters such as total gas pressure, partial pressures of reactive gases, deposition rate, temperature, and substrate material [16,29]. Though some researchers attributed the inclusion of more $O_2$ into the forming films, during deposition to high $N_2$ ratios in the gas flow [12,29], others have attributed that to low deposition rates rather than the $N_2$% age [3]. According to the results of this work, the previous EDX and XRD results suggest that the selected parameters in this work ($N_2$ flow rate and purging pressure) have favored the formation of a highly strained $TiN_x$ structure, which changed to a more stable TiN or $TiN_xO_y$ thin layer, after annealing at 400 °C in vacuum and air; respectively. The trends shown by Table 1 indicate no changes in the stoichiometry of the TiN layer before and after vacuum annealing, while changes have occurred on the structural parameters such as the microstrain and the grain size, which resulted in the observed changes on light absorbance. The absence of an $O_2$ peak for vacuum-annealed samples would also support this, though not a strong evidence. However, there is evidence that after annealing in air at 400 °C $N_2$ atoms are replaced by $O_2$ atoms. Compared with previous work [16,24], the optical absorbance of 95.4% achieved in this work for the air-annealed TiN was not reported before for TiN neither in as-deposited or annealed states. The enhancement in the optical absorbance after annealing (contradicting with previous work [16,24]) is attributed to the control of the deposition parameters (N/N + Ar ratio and pressure) and the annealing temperature. This work shows that control of deposition parameters to obtain $TiN_x$ rather than stoichiometric TiN is favorable for the optical absorbance properties. The annealing temperature should be also maintained below the temperatures of complete oxidation. Similarly, results contradicting with others were reported by Nikhil et al. [25], who reported enhancement in electrical properties of TiN sputtered thin films after annealing. Air annealing of the overstoichiometric TiN resulted in similar composition as the $TiN_xO_y$ sputtered thin film, different microstructures and accordingly different surface roughness. The equiaxed globular microstructure and lower surface roughness of the 400 °C air-annealed TiN caused the enhancement in its optical absorbance compared to the $TiN_xO_y$ sputtered film. This suggests a more economic and safer route to produce $TiN_xO_y$ thin films by sputtering TiN thin film, followed by annealing.

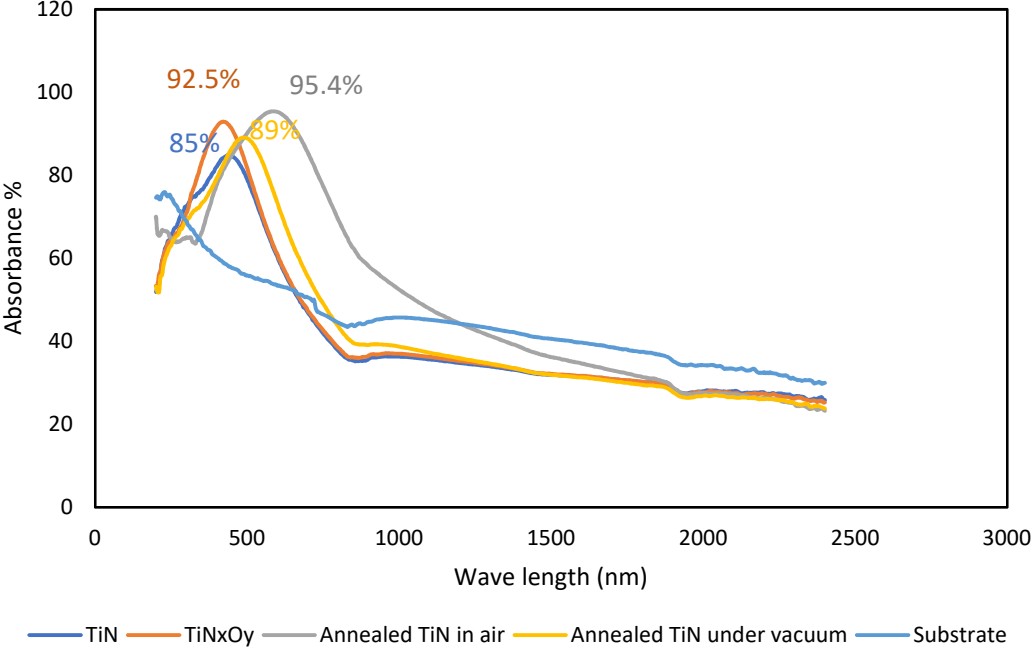

**Figure 4.** Absorbance of TiN, $TiN_xO_y$, annealed TiN in air, and vacuum.

The optical absorbance of the material obtained in this work, as shown in Figure 4, illustrates a red shift in the absorbance properties toward higher wavelengths for the annealed TiN in air and vacuum. This shift is larger for the annealed in air TiN. As shown from EDX results, and in Figure 2e,f and in Table 1, this red shift may be attributed to the increase in $O_2$ and C [36,37]. These results prove that TiN which is deposited under promoting N/Ti ratios in the range (1:1.43–1.66) or N compositional range of 65–70 at % is a good candidate for SSA applications, after vacuum and air annealing at 400 °C.

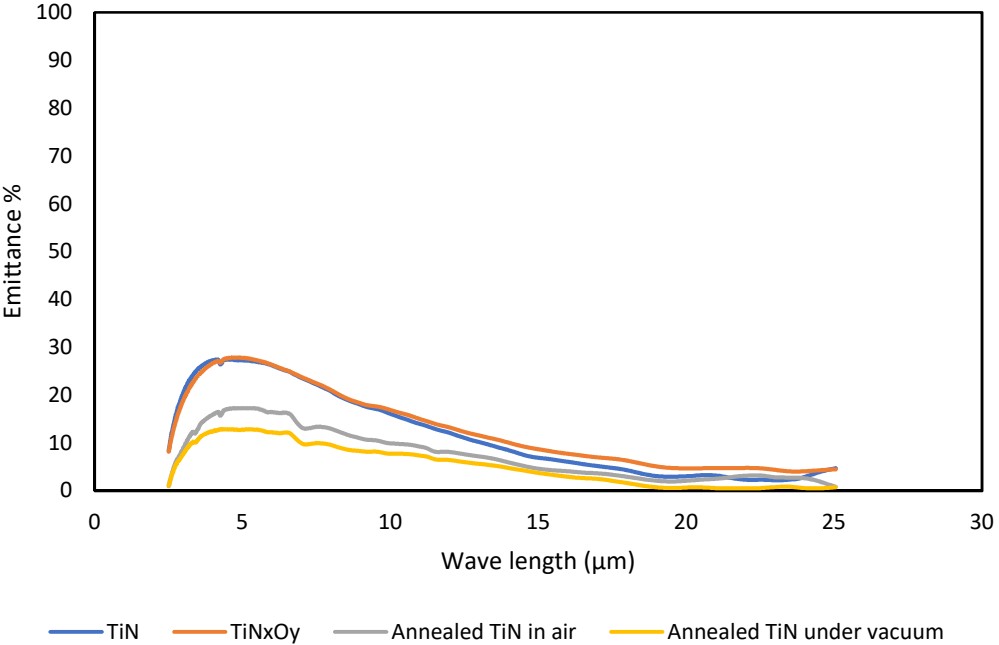

**Figure 5.** Emittance of TiN, $TiN_xO_y$, annealed TiN in air, and vacuum.

## 4. Conclusions

This work illustrates a comparison between the optical properties and the microstructure of $TiN_xO_y$, TiN, and annealed TiN at 400 °C (in air and in vacuum). The results prove that both thin films (TiN and $TiN_xO_y$) have comparable optical properties, and different structures, and microstructures. The maximum recorded optical absorbance for the as-deposited TiN thin film was about 85%. The optical absorbance was slightly higher (92.5%) in the case of as-deposited $TiN_xO_y$ thin film, while after annealing, the air-annealed TiN thin film showed the highest absorbance percentage (95.4%) and the vacuum-annealed showed the lowest emittance percentage (1%). This work has shown that changing the microstructure of the as-deposited TiN, by annealing, from nanometric-sized particles to globular-equiaxed nanoparticles has resulted in changes in the selective absorbance pattern of the thin film. It is also shown that $TiN_xO_y$ may be obtained from annealing TiN at 400 °C and the produced microstructure is completely different from the as-sputtered $TiN_xO_y$, the change from the columnar to equiaxed microstructure resulted in enhancement in the optical absorbance from 92.5% to 95.4%. All investigated thin films showed a similar trend of absorbance pattern with the same unstable profile. Since TiN is simpler and safer to manufacture, this work introduces vacuum- and air-annealed TiN thin film as a candidate absorber layer in SSA applications instead of direct sputtering of $TiN_xO_y$ thin films.

**Author Contributions:** Conceptualization, I.S.E.-M. and H.A.A.E.-F.; Methodology, H.A.A.E.-F.; Validation, M.H.S. and W.A.K.; Formal Analysis, I.S.E.-M. and H.A.A.E.-F.; Investigation, H.A.A.E.-F.; Data Curation, I.S.E.-M.; Writing-Original Draft Preparation, H.A.A.E.-F.; Writing-Review & Editing, I.S.E.-M. and M.H.S.; Visualization, M.H.S.; Supervision, I.S.E.-M.; Project Administration, M.H.S.; Funding Acquisition, I.S.E.-M. and M.H.S.

**Funding:** This research was funded by The Science & Technology Development Fund (STDF) of Egypt (No. 10663).

**Acknowledgments:** The authors thank the Center of Excellence, Nano Technology Center in Egypt, and the Science & Technology Development Fund (STDF) of Egypt Project (No. 10663).

**Conflicts of Interest:** The authors declare no conflict of interest.

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
