# Peer review of "Optical Properties and Microstructure of TiNxOy and TiN Thin Films before and after Annealing at Different Conditions"

_coatings, doi:10.3390/coatings9010022_

Round 1

Reviewer 1 Report

The authors present that optical properties and microstructure of TiNxOy and TiN thin films before and after annealing at different conditions. It reported that after air-annealing the optical absorbance of TiN exceeds that of TiNxOy. However, a lot of mistakes were showed in the manuscript.

1.          Many symbols are not shown properly in this manuscript. Please check whether the file is proper in your computer. For example, the chamber was pumped down to 10-4 Pa in line 100, and the Stefan-Boltzmann constant (5.6696x10-8 Wm-2K-4) in line 130.

2.          The SEM micrographs of the as deposited TiN, TiNxOy and annealed TiN at 400 in air and in vacuum thin films. What do you mean “400”?

3.          On the other hand, around 90% of solar energy is in the wavelength range of 0.3–1.5 μm.28 Therefore, the transition point (λc) is usually set at the wavelength of 1.5 μm [3]. What do you mean “28”?

As mentioned above, the authors must check the text carefully.

Author Response

Thanks are due to the reviewers for their valuable comments. All comments are considered and resulted the shown changes in the discussion of the work. The parts highlighted in red appear now in the text.

 Reviewer comment

Authors Reply

Many symbols are not shown properly in this manuscript. Please check   whether the file is proper in your computer. For example, the chamber was   pumped down to 10-4 Pa in line 100, and the Stefan-Boltzmann constant   (5.6696x10-8 Wm-2K-4) in line 130.

These mistakes are checked and corrected:

the chamber was pumped down to 10-4   Pa

Stefan-Boltzmann constant (5.6696x10-8   Wm-2K-4)

c1 =3.7405x108 Wμm4m-2   and c2=1.43879x104 μm K

and all the text has been revised for similar problems.

The SEM micrographs of the as deposited TiN, TiNxOy and annealed TiN   at 400 in air and in vacuum thin films. What do you mean “400”?

We are sorry for that typo mistake.

400 means 400°C (temperature degree) and corrected in text:

Figure 1 shows the SEM micrographs of the as deposited TiN, TiNxOy   and annealed TiN at 400°C in air and in vacuum   thin films.

On the other hand, around   90% of solar energy is in the wavelength range of 0.3–1.5 μm.28 Therefore,   the transition point (λc) is usually set at the wavelength of 1.5 μm [3].   What do you mean “28”?

We are sorry for that typo mistake.

28 is a reference but missing the brackets and is now corrected in   text:

On the other hand, around 90% of solar energy is in   the wavelength range of 0.3–1.5 μm [28]. Therefore,   the transition point (λc) is usually set at the wavelength of 1.5 μm [3].

Reviewer 2 Report

Authors have described the properties of TiNxOy and TiN thin films annealed in different ambient, the main goal of this manuscript as described in introduction is to provide further evidence for applicability of TiN instead of TiNxOy as a selective absorber. Authors have examined several structural and optical properties at different conditions. Some properties have necessity to be handled in rather detail and elaborately.

There are a few comments/questions which need to be addressed:

1)     In Table 1, compositions (at% or wt%) of Ti/N/O in all thin films showed very low percentages such as 2.51 at%. Would it be reasonable to assume that the rate of change of Fe/Ni/Cr in Fig. 2 had a greater impact than that of Ti/N/O?

2)     When authors would discuss the microstructures of the thin films, some properties of grain size, lattice constant, strain, and dislocation density should be described from XRD.

3)     Discussion of findings requires strengthening. Too much relevant works of literature are cited; however, the discussion should be more elaborated when differing from these of literature.

4)     How many samples of each type were tested? How many experiments were conducted? This has to be clearly mentioned and statistical errors should be incorporated in table 1.

5)     Why is there a difference between absorbance (Fig. 4) and emittance (Fig. 5) unlike Formula (1)?

6)     The figure number is very confused such as Figure 5a, Figure 3e, and Figure 3f. I can’t find those figures.

Author Response

Dear Reviewer

All replies are in submitted file.

Sincerely

Round 2

Reviewer 1 Report

Not enough results were disclosed in the manuscript because of these points listed below:

1. Compared to the previous study [16], no obvious result was showed in this manuscript.

2. The thickness was found to be 94 nm and 103 nm for TiN and TiNxOy, respectively. If the author can offer the cross-section image, it is easy to see the thickness.

3. From the SEM images, a lot of cracks were observed and it is not a continuous film. Will it affect the optical absorbance?

4. It is hard to define the grain size by the AFM images. The author should provide the TEM result.

Author Response

Thanks are due to the reviewers for their valuable comments. All comments are considered and resulted the shown changes in the discussion of the work. The parts highlighted in red appear now in the text.

1. Compared to the previous study [16], no obvious result was showed in this manuscript.

In this work the thin film was deposited on an opaque material, and only the absorbance in %age is available. This is the same as Ref 24. More emphasis on the results of this work is now illustrated in the manuscript.

2. The thickness was found to be 94 nm and 103 nm for TiN and TiNxOy, respectively. If the author can offer the cross-section image, it is easy to see the thickness.

A cross- section image is not available, due to the size and dimensions of the substrate, the samples could not fit vertically inside the SEM.

3. From the SEM images, a lot of cracks were observed, and it is not a continuous film.

Will it affect the optical absorbance?

It has been shown in the manuscript that these crack-like features in the as deposited microstructure exhibits well-defined large areas conforming to the original substrate grains with clear facets separated with large surface cavities along substrate original boundaries. It may affect the optical absorbance, but the scope of this work is investigating the effect of annealing. For comparison, the absorbance properties of the substrate are added to Fig. 4.

4. It is hard to define the grain size by the AFM images. The author should provide the TEM result.

TEM is not used for this work and no TEM results are available. Unfortunately, we cannot address this request, but grain size measurements by XRD have been added to the text.

Reviewer 2 Report

Authors have revised prior version of their manuscript. However, there remained several unsolved questions:

1)     In Table 1, compositions (at% or wt%) of Ti/N/O in all thin films showed very low percentages such as 2.51 at% when those of Fe/Mn/Cr from the substrate varied greatly in Fig. 2(a)-(g). How can the authors conclude that "their results do not affect the thin film”?

2)     Authors did not discuss some properties for the microstructures of the thin films, such as grain size from XRD, strain, and dislocation density in this revised version.

3)     There are too many citations of the previous literature without the authors’ own idea, discussion, and opinion differing from cited literature.

Author Response

Thanks are due to the reviewers for their valuable comments. All comments are considered and resulted the shown changes in the discussion of the work. The parts highlighted in red appear now in the text.

1.     In Table 1, compositions (at% or wt%) of Ti/N/O in all thin films showed very low percentages such as 2.51 at% when those of Fe/Mn/Cr from the substrate varied greatly in Fig. 2(a)-(g). How can the authors conclude that "their results do not affect the thin film”?

Answer:

The data presented in in Table 1 shows the EDX analysis results, which usually have a depth of resolution of 1-2 m m. This is the reason for the appearance of the substrate constituting elements in the EDX analysis. Also, the depth of resolution for Spectrophotometry and FTIR are in the range of 1200 nm. The effect of the substrate and the elements of the substrate alloy would be fixed across all prepared samples. Therefore, the authors think that the different values of Cr, Ni and Fe that appear in Table 1 are irrelevant to the measured optical properties.  For comparison, the absorbance properties of the substrate are added to Fig. 4.

2.     Authors did not discuss some properties for the microstructures of the thin films, such as grain size from XRD, strain, and dislocation density in this revised version.

The following is added to the text.

The XRD based grain size calculations have shown the following results.

The values of the mean grain size (crystallite size) and micro-strain obtained for the as-deposited and annealed thin films are summarized in Table 3. The broadening of the 111, 200, and 220 peaks was used to calculate the average grain size using Scherrer’s equation. It was found that the grain size of TiN thin film was 4 nm and grain size of TiNxOy thin film was 3.5 nm. The grain size for the TiN film annealed at 400°C in air was 6 nm and increased to 7 nm in the case of the film annealed at 400°C in vacuum. Light scattering and trapping at the junctions of tiny microcrystalline grains, can affect the absorbance.  Hence achieving a fine grain size is extremely important to achieve good absorbance. 

Table 3 Grain size and strain of as deposited TiN, annealed TiN in air and in vacuum, and TiNxOy thin films

TiNAnnealed TiN in airAnnealed TiN in vacuumTiNxOy
D ± nm 4673.5
ή × (103)6.44.84.556.5

The broadening of XRD peaks in nanocrystals is common and can also occur due to strain in the crystal or overlapping with other peaks. Peak broadening is normally seen in crystallites smaller than 100 nm.

The XRD results obtained in this work show that though the post-deposition annealing of the films did not cause significant variation in stoichiometry, but it affected the structural parameters such as micro-strain and crystallite size in a manner showing the effect of annealing on reducing the lattice strain and the defects resulting from the deposition conditions. This is also supported by the slight increase in Ti At% after vacuum annealing (as shown in Table 1) and the decrease in micro-strain after annealing as shown in Table 3.

3.     There are too many citations of the previous literature without the authors’ own idea, discussion, and opinion differing from cited literature.

A new richer discussion of the results is now added to the text, all highlighted in red for ease of follow up.

General additional issues

1.    English language is improved.

2.    Results and discussion improved.

Conclusions modified to describe the main results. 

Round 3

Reviewer 1 Report

It is a casual revision. Need rejection but giving a chance.

Reviewer 2 Report

The revised version of this manuscript was improved. This referee accepts for publication in its present form.